# Controlled Release Mechanism of Vancomycin from Double-Layer Poly-L-Lactic Acid-Coated Implants for Prevention of Bacterial Infection

**DOI:** 10.3390/polym14173493

**Published:** 2022-08-26

**Authors:** Papon Thamvasupong, Kwanchanok Viravaidya-Pasuwat

**Affiliations:** 1Department of Chemical Engineering, Faculty of Engineering, King Mongkut’s University of Technology Thonburi, 126 Pracha-Utid Rd., Bangkok 10140, Thailand; 2Biological Engineering Program, Faculty of Engineering, King Mongkut’s University of Technology Thonburi, 126 Pracha-Utid Rd., Bangkok 10140, Thailand

**Keywords:** PLLA coating, orthopedic coating, vancomycin, antibacterial coating, controlled release, double-layer coating, adhesive strength

## Abstract

Implantation failure due to bacterial infection incurs significant medical expenditure annually, and treatment tends to be complicated. This study proposes a method to prevent bacterial infection in implants using an antibiotic delivery system consisting of vancomycin loaded into poly-L-lactic acid (PLLA) matrices. A thin layer of this antibiotic-containing polymer was formed on stainless steel surfaces using a simple dip-coating method. SEM images of the polymeric layer revealed a honeycomb structure of the PLLA network with the entrapment of vancomycin molecules inside. In the in vitro release study, a rapid burst release was observed, followed by a sustained release of vancomycin for approximately 3 days. To extend the release time, a drug-free topcoat of PLLA was introduced to provide a diffusion resistance layer. As expected, the formulation with the drug-free topcoat exhibited a significant extension of the release time to approximately three weeks. Furthermore, the bonding strength between the double-layer polymer and the stainless steel substrate, which was an important property reflecting the quality of the coating, significantly increased compared to that of the single layer to the level that met the requirement for medical coating applications. The release profile of vancomycin from the double-layer PLLA film was best fitted with the Korsmeyer–Peppas model, indicating a combination of Fickian diffusion-controlled release and a polymer relaxation mechanism. More importantly, the double-layer vancomycin-PLLA coating exhibited antibacterial activity against *S. aureus*, as confirmed by the agar diffusion assay, the bacterial survival assay, and the inhibition of bacterial surface colonization without being toxic to normal cells (L929). Our results showed that the proposed antibiotic delivery system using the double-layer PLLA coating is a promising solution to prevent bacterial infection that may occur after orthopedic implantation.

## 1. Introduction

Implant-associated osteomyelitis is a common problem in orthopedic surgery and is caused by bacterial infection, mostly from the Gram-positive bacterium *Staphylococcus aureus* (*S. aureus*), leading to bone destruction and necrosis [1]. The standard treatment for infections caused by implants consists of the removal of the infected implant, debridement of damaged tissue, and long-term antibiotic administration [2]. However, systemic antibiotic therapy may not be effective, as the bacteria can adhere to the surface of the implant and form a biofilm, preventing antibiotics from penetrating the infected area and causing bacterial resistance toward antibiotic treatment [3]. Local antibiotic delivery using poly (methyl methacrylate) (PMMA) or bone cement was then suggested. Although PMMA bone cements containing antibiotics are commercially available, many surgeons still manually mix antibiotics into PMMA during a revision procedure [4]. Only a limited amount of antibiotics (approximately 10–15 wt%) is recommended to be loaded into PMMA cement, as it can significantly compromise the mechanical strength of PMMA cement [5]. Another major shortcoming of this method is its drug elution kinetics. An initial burst release of antibiotic drugs from PMMA cement was observed in the first 4 h, followed by a steep decrease until day 9 [6]. A burst release in drug delivery is undesirable, as it can lead to a high antibiotic concentration at the target site, which can cause tissue toxicity [7]. In addition, since PMMA is nonbiodegradable, a second surgery is required to remove the bone cement, which could impair healing and increase treatment costs as well as patient risk. Therefore, the development of prostheses with local antibiotic delivery in a controlled manner could aid in successful infection management.

Recently, antibiotic delivery systems for alloy medical devices have been developed to prevent bacterial infection. One approach is to permanently crosslink an antibiotic drug onto a metal surface of an implant. A study by Martin Rottman et al., in which vancomycin was tethered on titanium discs, showed that the tethered discs displayed satisfactory outcomes to inhibit bacterial growth [8]. However, since the drug molecules are permanently fixed to the metal surface, they are not able to diffuse to the adjacent tissues or to body fluids through the biofilm [9]. In addition, the antimicrobial activity of the active agents can be lost due to irreversible binding to bacteria, making the antibiotic surface spent [10].

The concept of antibiotic release coatings was proposed to reduce bacterial adhesion and avoid biofilm formation on implant surfaces. In this technology, antimicrobial compounds are mixed with polymeric materials (biodegradable or nonbiodegradable) and deposited on the surface of an implant by impregnation, physical adsorption, conjugation or complexation [11]. The clear advantage of this method is that the polymer used for coating can be tailored to control the release of a drug molecule at a desired rate and duration [12]. One of the most commonly used polymers in medical applications as an antibiotic carrier is polylactic acid-based polymer (PLA) because of its biodegradable properties and its approval by the FDA for use in medical devices [13]. PLA exists in three forms: poly(L-lactic acid) (PLLA), poly(D-lactic acid) (PDLA), and their racemic mixture, poly(DL-lactic acid) (PDLLA). While both are semi-crystalline, PLLA exhibits higher crystallinity than PDLA, leading to better chemical stability, more structural integrity and a slower degradation rate, which is suitable for use in drug delivery applications [14,15]. More importantly, PLLA degrades to L-lactic acid, which is the only form of lactic acid already produced by humans and mammals. On the other hand, D-lactic acid, the by-product of PDLA degradation, can cause health problems [16]. These properties make PLLA attractive for use with orthopedic implants due to its high tensile strength, which can withstand the mechanical loading applied in implant applications.

Although PLA-based coatings have already been studied by various groups to deliver a variety of antibiotic active agents [17,18], the isoforms of these coatings have not been specified. Different PLA stereoisomers possess different characteristics, specifically the degree of crystallinity, which directly affects their degradation rates, leading to entirely different drug release behaviors from the polymer matrix [19]. Another important characteristic of drug delivery coatings that has rarely been studied is the adhesion strength between the coatings and the substrates. Not only does the adhesion strength reveal the ability of the coating to withstand the process of implantation, it also controls the performance and outcome of any coated orthopedic implants [20]. The separation of the coating from the substrate, either by cracking or delamination, can lead to decreased levels of drug release and poor performance of the drug delivery system.

In this study, a controlled release system using double-layer PLLA for antibiotic delivery from a metal implant was developed. Vancomycin, a commonly used antibiotic to treat osteomyelitis [21], was selected as an antibiotic drug model in this study. A dip coating method was used to fabricate the drug-containing PLLA film with a drug-free topcoat on stainless steel with prolonged drug release. A mathematical model was developed to predict and understand the mechanism of vancomycin release from the double-layer PLLA system. In addition, the bonding strength between the polymer layer and the stainless steel substrate was evaluated. The antibiotic activities against *S. aureus* and the biocompatibility of the vancomycin-PLLA delivery system were also investigated.

## 2. Materials and Methods

### 2.1. Coating of Vancomycin-PLLA on Stainless Steel Plates

A 316 L stainless steel sheet purchased from a local supplier was cut to a size of 3 cm × 3 cm using an electrical discharge machining (EDM) wire cut. Abrasive scrubbing was employed to ensure a smooth surface, followed by immersion in 65% nitric acid at 60 ± 5 °C for 30 min.

Vancomycin hydrochloride powder (Siam Bheasach, Bangkok, Thailand) and PLLA (5 dL/g pellets, M.W. ~325,000–460,000), supplied by Polysciences, Warrington, PA, USA, were mixed at a ratio of 1:5 by weight using dichloromethane (Sigma-Aldrich, Singapore) as a solvent. The concentration of vancomycin was maintained at 10 mg/mL for all experiments. The completely mixed solution was used for coating immediately. The pretreated stainless steel plates were dipped vertically in the polymer solution using forceps. The thickness of the coating was controlled by an immersion duration of 10 s with 15 s of withdrawal time. The coated plates were then dried in a vacuum oven at 37 °C for 24 h. To determine the loading of vancomycin in the PLLA coating, stainless steel substrates were weighed before and after the coating procedure. The amount of vancomycin in the coating was calculated according to the formulation of the PLLA solution. To fabricate a drug-free topcoat, the vancomycin-polymer-coated plates were dipped into the PLLA solution without vancomycin as described previously. Afterwards, the plates were dried in a vacuum at 37 °C for an additional 24 h. A schematic diagram of the preparation procedure for the double-layer PLLA coating is presented in Appendix A Appendix A.

### 2.2. Characterization of the Vancomycin-Loaded PLLA Coating

#### 2.2.1. Morphology and Thickness

Both the surface morphologies and the cross-section of the coating were observed using a scanning electron microscope (SEM, JEOL, JSM-6610LV, USA). Image analysis software (ImageJ, NIH) was used to estimate the average pore diameter and thickness. The degree of uniformity was indicated by a coefficient of variance (%CV), as shown in Equation (1):(1)$$\%\mathrm{CV}=\frac{SD_{d}}{d_{\mathrm{ave}}}\times 100$$
where *d*_ave_ is the average thickness of the coating and *SD_d_* is the standard deviation of the thicknesses of the coating.

The dispersion of vancomycin in the PLLA matrix was investigated using elemental mapping of chlorine atoms, as chlorine was only present in vancomycin.

#### 2.2.2. Adhesive Strength between the Vancomycin-PLLA Layer and Stainless Steel Substrate

The adhesive strength of the coating was measured by a universal tensile testing machine (SHIMADZU, AG-X, Japan). The test method was modified from ASTM D5179-16, a standard test method for measuring the adhesion of organic coatings in the laboratory by the direct tensile method [22]. The one-sided-coated stainless steel plate was attached to the holder using Epoxy-2216 Gray. The samples were heated to 60 °C for 10 min and left at room temperature for at least 24 h to accelerate the curing process. The test was carried out at a pulling rate of 0.5 mm/min under dry conditions at room temperature and terminated at the point of coating separation.

### 2.3. In Vitro Release of Vancomycin

The coated plates were immersed in simulated body fluid (SBF) to simulate the human body’s condition and placed in a 37 °C incubator. The composition and preparation of SBF are described in Appendix A Appendix A. SBF solution was removed every hour and replenished with an equal volume of fresh SBF solution. The amount of vancomycin released was analyzed and fitted to various kinetic models.

The concentration of vancomycin was determined by the colorimetric method described by Fooks, J. R. et al. [23]. The sample containing vancomycin was treated with 5% sodium carbonate solution and 25% Folin–Ciocalteu reagent at a ratio of 2:2:1 by volume. Next, sonication was used to homogenize the solution. Color development occurred after the addition of the Folin–Ciocalteu reagent. Because the color intensity changed over time, the absorbance of the sample at a wavelength of 725 nm had to be measured two hours after the reaction for signal stability. The calibration curve between vancomycin concentration and its absorbance is shown in Appendix A Appendix A.

### 2.4. Antibacterial Activity

The methods to test the antibacterial activity of the polymeric controlled release system using a survival assay and an agar diffusion assay were modified from Ordikhani et al. [24]. The test samples were 3 m × 3 cm stainless steel plates coated with vancomycin-PLLA, drug-free PLLA as a negative control, and a 10 mg/mL vancomycin solution-immersed paper filter as a positive control. The surfaces of these samples were sterilized by 70% ethanol, according to the protocol by Graziano et al. [25] Briefly, 70% ethanol was applied directly on all the sample surfaces, with a period of contact of at least 10 s, before being wiped clean with a sterile cloth. This process was repeated 3 times to ensure that there was no contamination. For the survival assay, *S. aureus* (ATCC25923) was inoculated in nutrient broth at 37 °C overnight. Five hundred microliters of *S. aureus* bacterial suspension at 5.0 × 10^5^ CFU/mL was dropped on the coated plate samples and incubated at 37 °C. After 12 h of incubation, 100 μL of the exposed culture was serially diluted, plated on nutrient agar, and incubated overnight at 37 °C to determine the residual bacteria. The CFUs of the surviving bacteria were counted, and the bacterial reduction percentage was calculated, according to Equation (2).
(2)$$\%\mathrm{Reduction}=\frac{B-A}{B}\times 100$$
where *B* is the initial concentration of bacteria and *A* is the concentration of bacteria after 12 h of incubation.

In the agar diffusion assay, *S. aureus* bacteria at a density of 3.8 × 10^10^ CFU/mL were applied uniformly to an agar plate and further incubated for two hours. The coated plate samples, as described previously, were placed on the surface of each plate. After 24 h at 37 °C incubation, the average inhibition area was calculated with three replications. The equivalent diameter (*D_eq_*) of inhibition was calculated as in Equation (3).
(3)$$D_{eq}=\sqrt{\frac{\pi \cdot Area}{4}}$$

The relative percentage of inhibition was calculated using Equation (4), where $D_{eq^{+}}$ is the equivalent diameter of inhibition of the positive control. In this study, the control was a 10 mg/mL vancomycin solution-immersed paper filter.
(4)$$\mathrm{Relative}\;\mathrm{Percentage}\;\mathrm{of}\;\mathrm{Inhibition}=\frac{D_{eq,sample}}{D_{eq,+}}\times 100$$

Another antibacterial activity was crystal violet staining to study bacterial attachment to substrate surfaces. *S. aureus* was grown overnight in trypticase soy broth (TSB) at 37 °C and 200 rpm. Bacterial cells were collected by centrifugation at 10,000× *g* for three minutes and resuspended in normal saline solution (0.85% NaCl) to a final cell concentration of 10^8^ CFU/mL (OD_600_ = 0.08–0.12). The cell suspensions were 100-fold diluted in TSB to a final cell concentration of 10^6^ CFU/mL. Next, the broth was poured into Petri dishes containing samples and incubated at 37 °C for 24 h without shaking. The assay was performed in parallel with negative controls, which were an uncoated stainless steel plate and a PLLA-coated stainless steel plate without vancomycin. After incubation, the bacterial culture was removed by pipetting, and the samples were rinsed with PBS to remove planktonic cells. The contact surfaces were stained with 0.1% aqueous crystal violet for 30 min at room temperature. Bacterial colonization was determined by the presence of the violet color on the contact surface.

### 2.5. Cytotoxicity Study

The biocompatibility of the vancomycin-loaded PLLA coating was evaluated following the protocol of ISO-10993-5 [26] using the MTT assay. In this study, 3 × 3 cm^2^ plain stainless steel plates, PLLA with and without vancomycin coated on stainless steel plates, were sterilized using 70% ethanol, as previously described, before being incubated in Dulbecco’s modified Eagle’s medium (DMEM) with low glucose (Sigma Aldrich, St. Louis, MO, USA) for 24 h at 37 °C. L929 mouse fibroblast cells (Mouse C3H/An) were seeded on a 96-well plate at a density of 10,000 cells/well and allowed to grow until confluence overnight. Afterwards, 100 μL of the media previously incubated with the test samples was added to monolayers of confluent L929 cells and further incubated for another 24 h. The cells cultured in growth medium served as a positive control, while the cells treated with 1% phenol served as a negative control. After 24 h of incubation, the spent medium was discarded and replaced with MTT solution according to the manufacturer’s instructions (Thermo Fisher Scientific, Waltham, MA, USA). The absorbance of dissolved formazan salt was measured at 570 nm (Infinite^®^ 200 Tecan, Grödig, Austria). The cell viability is presented as a percentage of the control cells, as shown in Equation (5).
(5)$$Cell\;Viability\;\left(\%\right)=\frac{Absorbance\;at\;570\;\mathrm{nm}\;of\;the\;treated\;sample}{Absorbance\;at\;570\;\mathrm{nm}\;of\;the\;control\;sample}\times 100$$

### 2.6. Statistical Analysis

Data are presented as the mean ± standard deviation (SD). All results were analyzed by an unpaired *t* test. A statistically significant difference was defined at a 95% confidence level (*p* < 0.05).

## 3. Results and Discussion

### 3.1. Characterization of the Vancomycin-Loaded PLLA Coating

Fabrication of polymeric coatings can be achieved using various techniques, including dip coating, spin coating, spray coating, and solvent casting. Among these methods, dip coating is the most commonly used in laboratories and industries because of its simplicity, cost-effectiveness, reliability and reproducibility [27]. This method is especially useful for researchers to quickly and inexpensively optimize processing parameters for their studies. The thickness of the films fabricated by dip coating can be controlled with the duration of immersion, speed of withdrawal, and rheological properties of the coating solution [28].

The PLLA coating in this study appeared opaque and was uniformly distributed on the surface of the stainless steel. According to Figure 1, the SEM images of the PLLA coating on stainless steel without vancomycin (Figure 1A,B) show a honeycomb network of the polymer with pore sizes of 14.85 ± 3.07 μm. The formation of the honeycomb structure, previously explained by Escalé, L. et al. [29], began with the rapid evaporation of the organic solvent causing a quick temperature drop at the surface under the dew point, leading to fast condensation of water vapor into water droplets on the cold surface. The water droplets were, then, self-organized into an ordered hexagonal lattice. In a solution with a thermal gradient, the convection flow promoted the regular stacking of the water droplets. After the evaporation of both organic solvent and water droplets was completed, a polymer film was formed with an organized pattern of small pores, which were previously occupied by water droplets. Factors influencing the formation and morphology of a honeycomb structure include the relative humidity, volatility of solvents, air velocity, and polymer concentration [30]. For example, the pore size can be decreased with an increase in the polymer concentration due to less space for water droplets to occupy, while a low air velocity results in a larger pore diameter.

The cross-sectional view revealed that the polymer had uniform thickness (Figure 1C), as confirmed by %CV in Table 1. When vancomycin was incorporated into the system, uniformly distributed drug particles were observed on the polymer surface (Figure 1D). When focusing on each drug particle, it was found that the particles did not incorporate themselves into the polymer network but rather were trapped inside the polymer layer (Figure 1E). This was confirmed by a cross-sectional view in which the vancomycin molecules filled in the space within the PLLA network structure, resulting in a denser matrix (Figure 1F). The porous network of PLLA remained the same, with an average pore size of 12.43 ± 4.08 μm. Judging from a cross-sectional view (Figure 1C,F), vancomycin clearly did not affect the average thickness of the coating, but the uniformity of the coating’s thickness decreased, which was indicated by an increase in the coefficient of variance (Table 1). As the porous network of PLLA remained unchanged, it indicated that vancomycin did not interfere with the microscopic structure of the polymer. This was possibly because of the immiscibility of the hydrophobic PLLA and hydrophilic vancomycin molecules [31]. One of the important factors for long-term drug release is the hydrophobicity of the matrix, which provides a significantly slow degradation rate [32]. The loading of vancomycin into the PLLA coating was measured to be 0.57 ± 0.07 mg/cm^2^. The amount of vancomycin loaded in the PLLA coating in this study was consistent with other studies using different polymer matrices, which demonstrated successful outcomes in treating bacterial infection [33,34].

The SEM images of the PLLA-vancomycin layer with a drug-free topcoat (Figure 2A) show a morphology similar to that of the coating of the PLLA network (Figure 1B). The drug particles were not easily seen from the top view due to the topcoat. In the cross-sectional view, a clear interface between the base layer and the topcoat could not be seen, possibly because the coating redissolved into the solvent during the second coating process, resulting in no clear interface between the two layers (Figure 2B). The thickness of the coating was found to be 54.09 ± 3.46 μm, which consisted of 41.35 ± 2.53 μm of the base vancomycin-PLLA layer and 12.74 ± 0.93 μm of the drug-free PLLA topcoat. According to the %CV, adding the drug-free topcoat remarkably improved the thickness uniformity from that of the PLLA containing vancomycin coating. 

The adhesive bond strength between the coating layer and the substrate is a major consideration in practice. The detachment of the coating from the substrate incurs unpleasant effects on the implants and the surrounding tissue [35,36]. The adhesion strengths between the coating and the stainless steel substrate determined by the direct tensile method are reported in Table 2. The bonding strength between the PLLA coating and a stainless steel substrate was 1.80 ± 0.49 MPa. The adhesion strength between the PLLA coating loaded with vancomycin and a stainless steel substrate was 1.87 ± 0.45 MPa, which was not significantly different from that of the PLLA coating without vancomycin. The bond strength between the PLLA layer and the stainless steel substrate in this study was of the same level as the bond strength between PLLA and metallic magnesium [37,38]. However, the double-layer PLLA coating with a topcoat exhibited a considerably higher bonding strength, which exceeded the epoxy strength (approximately 5 MPa). This could be explained by the longer processing time of the coating. Similar to the study by Morris B., an increase in the processing time reduced the stress within the coating layer, leading to a thicker film and stronger bond strength [39,40]. Although there is currently no ISO requirement for orthopedic implant coatings, the requirement for dental implant coatings can be used as a reference. According to ISO 10477, the adhesive bond strength at the interface between biocomposites and substrate should be greater than 5 MPa [41]. Only the adhesive bond strength of the vancomycin-PLLA coating with a topcoat met the ISO requirement. The topcoat layer clearly produced a stronger and more durable bond with the metal substrate, leading to the long-term reliability of the coating in implant applications.

### 3.2. Release Profile and Release Mechanism of Vancomycin from the PLLA Coating

The in vitro cumulative release of vancomycin from PLLA-coated stainless steel was investigated for up to 72 h (Figure 3A). The release profile of vancomycin shows a sudden burst release of approximately 95% of the total release in the first 12 h. After 72 h, there was no noticeable amount of vancomycin released into the receiving medium. The cumulative amount of vancomycin released from the system was calculated to be approximately 55% of the amount of vancomycin loaded. To investigate the kinetics of the drug release, several theoretical and empirical models were used to fit the experimental release data up to 24 h, as the amount of vancomycin release after 24 h was considered negligible. Table 3 shows the equations, rate constants (*k*) and exponents (*n*) of several selected kinetic models with their coefficients of determination (*R*^2^) to evaluate the goodness of fit of different models. The release data of vancomycin from the PLLA coating fit best with the Korsmeyer–Peppas kinetic model, with the highest coefficient of determination (*R*^2^) close to 0.99. Using the Korsmeyer–Peppas model, the initial release rate of vancomycin during the burst release period in the first six hours was calculated to be 1.6 mg/h with a calculated release exponent (*n*) of 0.21.

A drug-free PLLA topcoat was added to the existing vancomycin-PLLA coating to extend the drug release, as shown in Figure 3B. No initial burst release was observed from the double-layer coating. In the first 2 days, the initial release rate was reduced from 1600 μg/h in the single-layer vancomycin-PLLA coating to 50 μg/h with another layer of drug-free PLLA topcoat (double layer), which was an approximately 95% reduction. Furthermore, the release duration was remarkably extended to longer than 20 days. Postoperative bacterial infection leading to acute osteomyelitis typically occurs within 2 weeks [42]. The administration of antibiotics was recommended for the first three weeks after the surgery [43]. The current vancomycin release duration would be able to treat an early bacterial infection, which could subsequently prevent osteomyelitis. 

Elemental mapping of chlorine was performed to track the location of the vancomycin molecules and allow for the observation of vancomycin dispersion in the polymer matrix before and after the release experiment. Figure 4A displays the uniform dispersion of vancomycin in the polymer system before the in vitro release study. The cross-sectional view confirmed that the vancomycin molecules were entrapped under the layer of the PLLA matrix (Figure 4B). After immersion in the release medium for three weeks, chlorine element mapping illustrated a significant change in the vancomycin density in the polymer matrix (Figure 4C). Significantly less vancomycin was observed in the PLLA layer after three weeks, and the vancomycin molecules were mostly located near the surface of the coating.

To further investigate the release mechanism of vancomycin from the PLLA layer with the drug-free topcoat, the experimental data from Figure 3B were fitted with the same release kinetic models. According to Table 3, based on the coefficient of determination (*R*^2^), the experimental data would fit best with the Korsmeyer–Peppas model with an exponent of 0.54 (*n* = 0.54).

In the Korsmeyer–Peppas kinetic model, the exponent *n* is used to characterize the mechanism of drug release. For a thin film, *n* < 0.5 corresponds to a Fickian diffusion mechanism, 0.5 < *n* < 1 to non-Fickian transport, and *n* > 1 to case II transport or zero order release [44]. Since the release exponent of the formulation without a topcoat was less than 0.5, the release mechanism of vancomycin from the PLLA coating was thought to be Fickian diffusion, in which the release of an active compound depended on the drug content in the system. At the earlier stage, the vancomycin content was high, leading to the presence of burst release. The release rate was significantly lower after the reduction in the vancomycin content in the system. As shown by element mapping of chlorine in Figure 4, it is believed that the movement of the vancomycin molecules was controlled by the concentration gradient driving force [40]. This phenomenon implied that vancomycin diffused through the polymer matrix toward the surface of the PLLA coating and was released into the receiving medium.

Typically, a burst release in a drug delivery system is undesirable and cannot be used for a long-term release scenario, not to mention the toxic side effects due to the high initial concentration of a drug, leading to a decrease in the efficiency of the drug delivery system. In this study, the initial burst release of vancomycin from the PLLA layer was observed in the first 12 h, resulting in the released concentration being far above its minimum inhibitory concentration (MIC), which was approximately 2 μg/mL [45]. Too high a concentration of an antibiotic drug in the first twelve hours may be toxic to human cells [46]. Therefore, the release rate of vancomycin must be reduced to achieve the sustained drug delivery goal. Because the release mechanism is diffusion controlled, the release rate can be reduced by adding mass transfer resistance. One method to increase the mass transfer resistance is to fabricate a drug-free barrier, increasing the diffusion distance. In a previous study, the release kinetics for two different formulations of sirolimus drug-eluting stents showed a remarkable reduction in the release rate by adding a drug-free topcoat layer [47]. Another method used to reduce the burst release effect is the crosslinking of the polymer layer using crosslinking agents, which results in the creation of additional diffusional resistance [48]. The crosslinking agents are normally cytotoxic and usually need further validation of safety. To avoid using toxic crosslinking agents, heat treatment or ultraviolet radiation are alternatives [49]. However, high temperatures and exposure to radiation might cause some antibiotics to become inactive or degraded [50,51]. On the other hand, the topcoat method was able to provide the same level of safety, but the release mechanism tended to be more complicated. The release kinetics could be based on more than one mechanism [52]; for example, the combination of release mechanisms, including Fickian diffusion control and polymer relaxation control, was observed [53].

In this study, a drug-free PLLA topcoat was applied to the existing vancomycin-loaded PLLA layer to provide an additional mass transfer resistance layer that was believed to be able to reduce the release of vancomycin during the burst period and extend the release of vancomycin in a controlled manner. By adding the drug-free topcoat, the diffusion distance increased, leading to more resistance to mass transfer. As a result, a lower amount of vancomycin was delivered, leading to a lower release rate but longer release duration as a consequence of the resistant layer provided by the topcoat. The *n* value (0.54) indicates that the vancomycin delivery system was possibly controlled by two mechanisms, including both Fickian diffusion-controlled and polymer relaxation-controlled mechanisms. To further understand the release mechanism, the Peppas and Sahlin equation [48], a modification of the Korsmeyer–Peppas equation, as shown in Equation (6), was applied.
(6)$$\frac{M_{t}}{M_{\infty }}=k_{1}t^{m}+k_{2}t^{2m}$$
where *k*_1_ and *k*_2_ are the Fickian contribution coefficient and relaxation contribution coefficient, respectively. The first term of the exponent *m* of Equation (6) represents the release due to the Fickian diffusion mechanism, while the second term of the exponent 2*m* represents the release due to the polymer relaxation mechanism. By fitting the release profile of vancomycin from the PLLA layer with drug-free topcoat, *k*_1_ and *k*_2_ were found to be 0.058 h^−0.5^ and 0.001 h^−1^, respectively, with a release exponent (*m*) of 0.5 and *R*^2^ of 0.9954. These coefficients, *k*_1_ and *k*_2_, were then used to calculate the ratio of relaxational (*R*) over Fickian (*F*) contributions as a function of time, as shown in Equation (7) [48].
(7)$$\frac{R}{F}=\frac{k_{2}}{k_{1}}t^{m}$$

The Peppas–Sahlin model shows that the ratio of relaxation over the Fickian diffusion contributions increased over time, indicating that the effect of polymer relaxation became more pronounced on controlling the release of vancomycin as the time increased (Figure 5). However, the release of vancomycin from the system was still largely controlled by Fickian diffusion even after 16 days, as the ratio was still less than 0.5. According to the Peppas–Sahlin model, 98.30% of vancomycin release in the first hour was the result of Fickian diffusion, and the rest (1.70%) was from the polymer relaxation mechanism, while at later times, a lower amount of vancomycin (73.55%) was released by Fickian diffusion, and approximately 26.45% of vancomycin was released because of polymer relaxation. Fickian diffusion showed a smaller effect on vancomycin release at later times, possibly because the vancomycin content in the matrix was lower than that at earlier time points, resulting in a lower concentration gradient for diffusion. Additionally, previous studies revealed that PLLA tended to be slowly swollen and degraded after water molecules diffused into the matrix. The incorporation of drug particles in PLLA could also lead to an increase in the swelling index [54,55]. Therefore, our PLLA delivery system might have been swollen slightly after being incubated in aqueous solution for a few weeks. Compared with the formulation without the topcoat, the polymer was not exposed to the receiving medium long enough to be swollen. As a result, the mechanism of vancomycin release was expected to be only Fickian diffusion-controlled.

### 3.3. Antibacterial Activity of Vancomycin-Loaded PLLA

According to the survival assay results in Table 4, the vancomycin-PLLA coating with and without a drug-free topcoat was able to reduce the *S. aureus* population by approximately 81.77% and 91.77%, respectively. The PLLA coating without vancomycin, which served as a negative control, did not show any antibacterial activities against *S. aureus*, while the positive control, which was vancomycin-immersed filter paper, could completely eliminate bacteria from the system. An agar diffusion assay was also performed to confirm the antibacterial activities of the proposed vancomycin delivery system (Table 4 and Figure 6). Similarly, the vancomycin-PLLA coating without the topcoat gave a larger inhibition zone of 19.71 cm^2^ (or *D*_eq_ of 3.93 cm), while the inhibition area of the vancomycin-PLLA coating with the drug-free topcoat was estimated to be 12.97 cm^2^ (or *D*_eq_ of 3.19 cm). As expected, there was no inhibition zone for the negative control, but the positive control provided the largest inhibition zone of 24.56 cm^2^ (or D_eq_ of 4.39 cm). Compared to the control, the relative percentages of inhibition were calculated to be 72.67% and 89.52% for the vancomycin-PLLA coating with and without a topcoat, respectively. The presence of bacterial colonization, as indicated by crystal violet staining was observed on the stainless steel substrate (Figure 7A) and the PLLA coating without vancomycin (Figure 7B). The violet color was clearly visible on both samples, which represented the presence of the *S. aureus* culture. In contrast, the vancomycin-PLLA coating with and without a topcoat showed colorless samples, indicating no bacterial attachment on either surface (Figure 7C,D). Our results confirmed the antibacterial activity of the proposed formulations and that the concentration of vancomycin released from the system was sufficient to prevent bacterial surface attachment and growth on the material surfaces.

### 3.4. Cytotoxicity of the Vancomycin-PLLA Delivery System

The cytotoxicity of the vancomycin-PLLA delivery system was evaluated using an MTT assay, as shown in Figure 8A. Mouse fibroblast cells (L929) were exposed to the extracted media from plain stainless steel, PLLA coating without vancomycin, vancomycin-PLLA coating on stainless steel substrates, and 1% phenol (a negative control) for 24 h. As shown in Figure 8, the cell viability of L929 cells incubated with the spent medium from the stainless steel substrate was the highest at 97.35 ± 2.17%, while a slightly lower cell viability of L929 of approximately 90% was observed when the cells were incubated with the spent medium from the stainless steel substrates coated with drug-free PLLA and vancomycin-PLLA. According to ISO 10993-5: Biological evaluation of medical devices—Part 5, which stated that any materials with cell viability greater than 70% are considered noncytotoxic, our results showed that the cell viability of the PLLA coating with and without vancomycin was well above 70%. The morphology of L929 cells also confirmed that the extracts of all test samples were nontoxic (Figure 8B). The cells exposed to the extracts had normal fibroblast-like morphology and were well spread, similar to that of the cells maintained in the culture medium. Therefore, both quantitative and qualitative cytotoxicity assessments indicated that the vancomycin-loaded PLLA delivery system was not cytotoxic and could be considered safe for use in future applications.

There has been considerable interest in coating orthopedic implants with antibiotic impregnated PMMA bone cement, which has already been approved by the FDA [56]. Coating of antibiotic-PMMA on implants is usually carried out using a molding technique, resulting in a film with a thickness between 1–3 mm [56]. The release characteristic of PMMA consists of an initial burst release, followed by a decrease in the drug release to a level below its therapeutic range [6]. The low efficiency of the antibiotic-PMMA coating for local drug delivery is the main barrier for its clinical use. Another complication arising from the use of PMMA-coated implants is implant-PMMA debonding, which is reported to occur in approximately 10–30% of cases [56]. In this proof-of-concept study, PLLA was shown to represent a possible alternative solution, as it has been demonstrated to be able to deliver an antibacterial agent in controlled and sustained manners and is effective against *S. aureus*, the main cause of osteomyelitis. However, more studies are needed to thoroughly evaluate the potential use of this delivery system in preclinical and clinical settings. 

## 4. Conclusions

In this study, we successfully developed a vancomycin delivery system using PLLA coated on stainless steel. The PLLA film showed a honeycomb structure with a thickness of approximately 44 μm. After incorporation with vancomycin, the structure of the polymer film changed slightly, with vancomycin molecules trapped inside the polymer network, resulting in nonuniformity of the coating thickness. However, the thickness of the PLLA layer after vancomycin loading remained the same, leading to no change in the bonding strength between the coating and the stainless steel substrate. In the in vitro release experiment, the burst release of vancomycin was observed at an earlier time, and the sustained release was present afterwards. The release period of vancomycin was approximately three days for the single-layer coating. The release kinetics of vancomycin from the system fit best with the Korsmeyer–Peppas model, indicating that vancomycin release was dominantly controlled by Fickian diffusion. To solve the problem of undesirable burst release and to extend the release of vancomycin, a drug-free layer of PLLA was added to the existing vancomycin-loaded PLLA layer, acting as a diffusion resistance layer. With the drug-free topcoat, the bonding strength between the coating and the substrate significantly increased, indicating a stronger and more durable bond, which would be desirable in real applications. Moreover, the new system was able to extend the release period of vancomycin for more than 20 days. Although the mechanism of vancomycin release remained primarily controlled by Fickian diffusion, the contribution of the polymer relaxation to the release mechanism became more pronounced at later time points. The antibacterial assays confirmed that vancomycin embedded in a PLLA coating with and without a drug-free topcoat had the ability to inhibit the growth of bacteria without being cytotoxic to normal cells. Thus, the double-layer PLLA-vancomycin coating may represent a promising approach that can possibly prevent bacterial infection associated with implants.

## Figures and Tables

**Figure 1 polymers-14-03493-f001:**
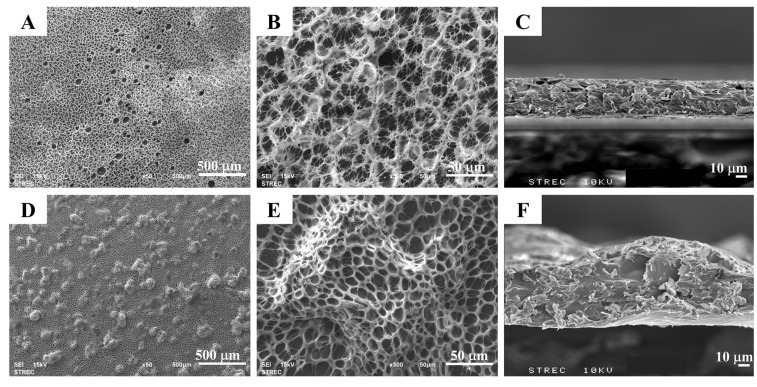
SEM images of the PLLA-coated layer: top view at (**A**) 50× and (**B**) 500×, and (**C**) cross-sectional view at 500× (the scale bars represent 500 μ, 50, and 10 μm, respectively). SEM images for the PLLA-coated layer containing vancomycin: top view at (**D**) 50× and (**E**) 500×, and (**F**) cross-sectional view at 500× (the scale bars represent 500, 50, and 10 μm, respectively).

**Figure 2 polymers-14-03493-f002:**
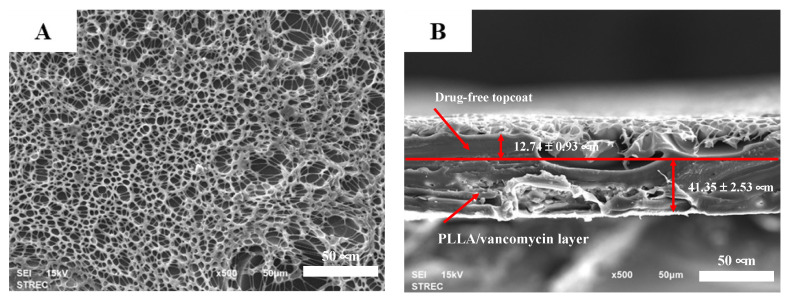
SEM images of the PLLA layer containing vancomycin with the drug-free topcoat: (**A**) top view at 500×, and (**B**) cross-sectional view at 500× (scale bars represent 50 μm).

**Figure 3 polymers-14-03493-f003:**
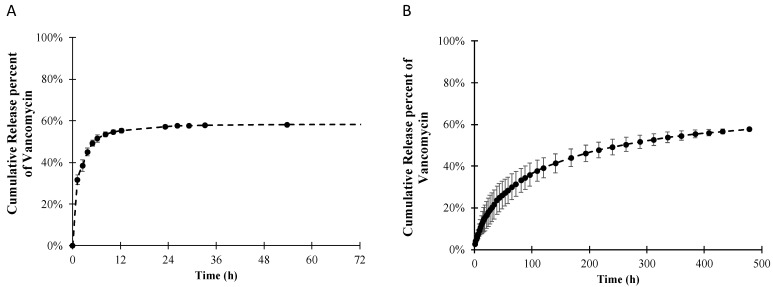
Release profiles of vancomycin from (**A**) a single PLLA layer and (**B**) double layer of PLLA with a drug-free topcoat.

**Figure 4 polymers-14-03493-f004:**
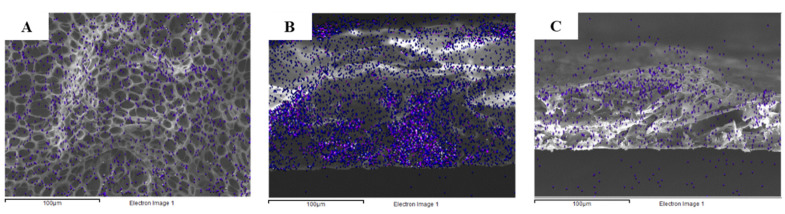
Chlorine elemental mapping of the PLLA layer containing vancomycin before the in vitro release experiment: (**A**) top view and (**B**) cross-sectional view, and (**C**) the cross-sectional view of PLLA containing vancomycin after 3 weeks in SBF (the scale bars represent 100 μm).

**Figure 5 polymers-14-03493-f005:**
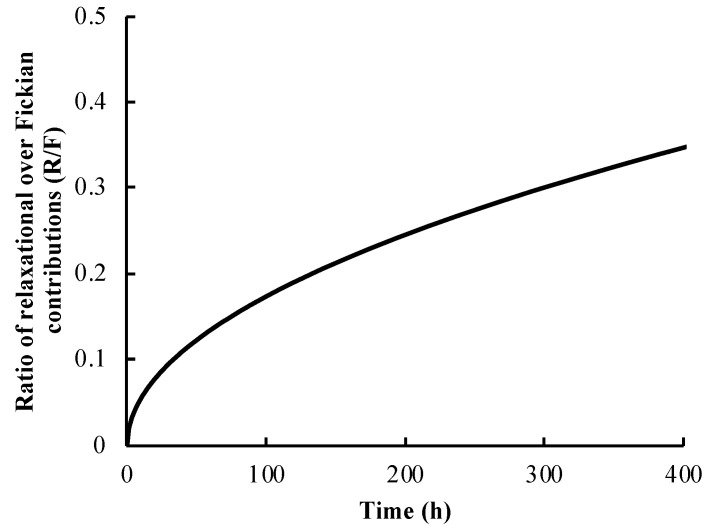
The ratio of relaxation over the Fickian contributions from the Peppas–Sahlin model for the vancomycin-PLLA coating with a drug-free topcoat.

**Figure 6 polymers-14-03493-f006:**
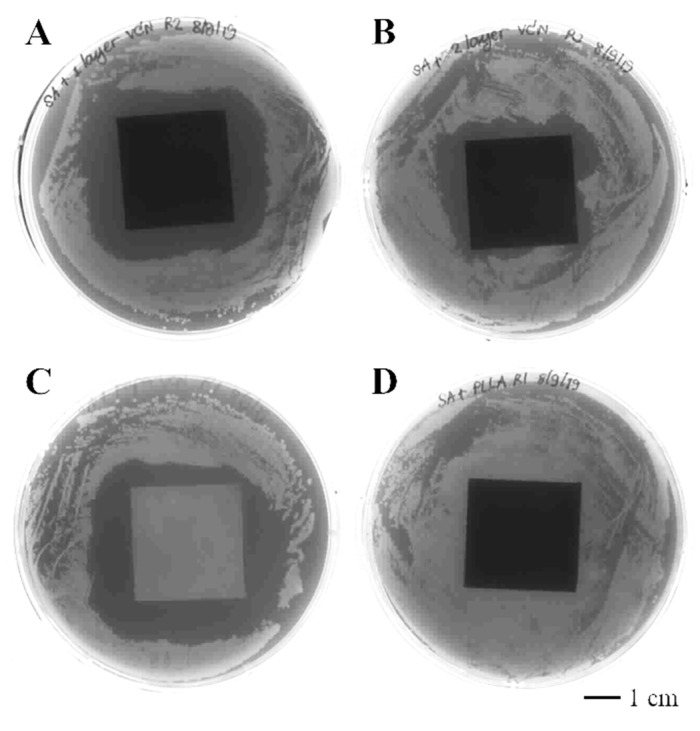
Inhibition zones of (**A**) vancomycin-loaded PLLA-coated plate without the drug-free topcoat, (**B**) vancomycin-loaded PLLA-coated plate with the drug-free topcoat, (**C**) 10 mg/mg vancomycin-loaded PVDF membrane as a positive control, and (**D**) PLLA-coated plate without vancomycin as a negative control.

**Figure 7 polymers-14-03493-f007:**
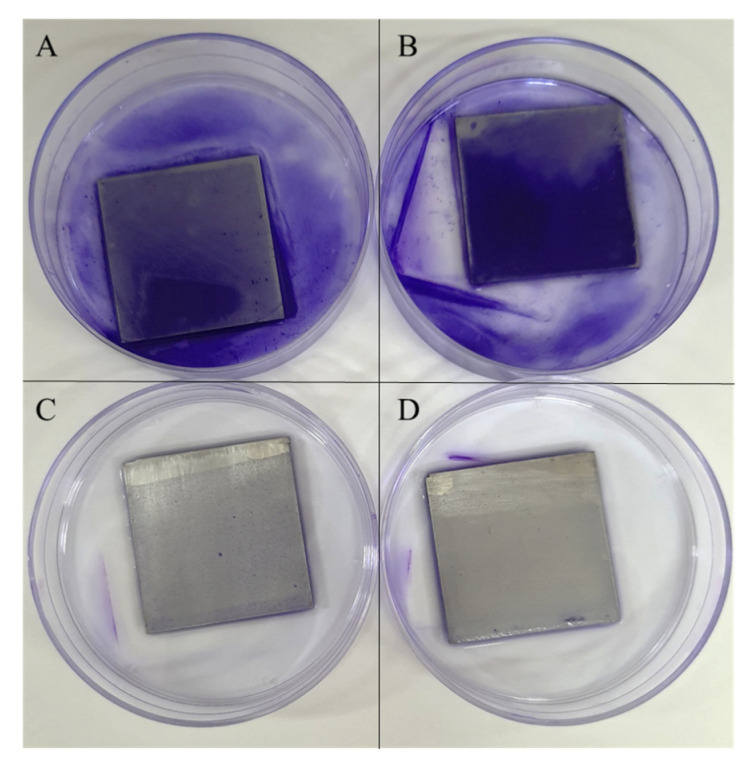
Crystal violet staining assay (**A**) stainless steel (**B**) PLLA-coated stainless steel (**C**) vancomycin loaded PLLA-coated stainless steel, and (**D**) vancomycin loaded PLLA-coated stainless steel with a drug-free topcoat.

**Figure 8 polymers-14-03493-f008:**
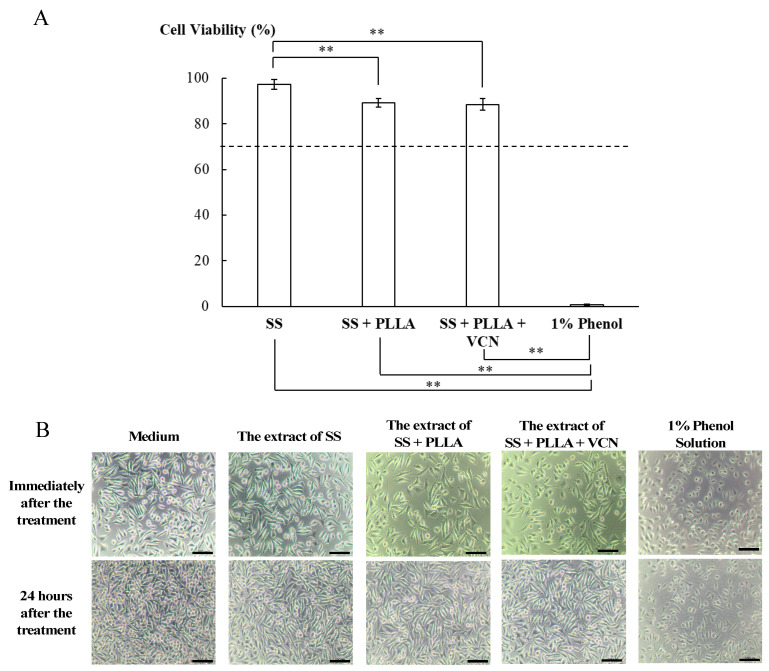
(**A**) Cell viability of different substrates including stainless steel (SS) as a negative control, PLLA coated on stainless steel (SS + PLLA), PLLA containing vancomycin coated on stainless steel (SS + PLLA + VCN), and 1% phenol as a positive control (** indicates *p* < 0.001). (**B**) Morphology of L929 cells after exposure to its culture medium (control), the extract of stainless steel (SS), the extract of PLLA coated on stainless steel (SS + PLLA), the extract of PLLA containing vancomycin coated on stainless steel (SS + PLLA + VCN) and the culture medium containing 1% phenol.

**Table 1 polymers-14-03493-t001:** Thickness of the PLLA films and their coefficients of variance.

Formulation	Average Thickness (μm)	Coefficient of Variance (%)
PLLA coating	44.57 ± 2.51	5.63
Vancomycin-PLLA coating	44.43 ± 10.13	22.79
Vancomycin-PLLA coating with a drug-free topcoat	54.09 ± 3.46	6.39

**Table 2 polymers-14-03493-t002:** Bonding strength between the polymeric coatings and stainless steel substrates using a direct tensile method.

Sample	Bonding Strength (MPa)
PLLA coating and stainless steel	1.80 ± 0.49
Vancomycin-PLLA coating and stainless steel	1.87 ± 0.45
Vancomycin-PLLA coating with a topcoat and stainless steel	>5 MPa

**Table 3 polymers-14-03493-t003:** Equations and parameters of the theoretical and empirical kinetics models for the vancomycin-PLLA coatings: single layer and double layer.

Model	Equation	Vancomycin-PLLA Coating			Vancomycin-PLLA Coating with a Drug-Free Topcoat		
		*K*	*n*	*R^2^*	*k*	*n*	*R^2^*
Zero order	$\frac{M_{t}}{M_{\infty }}=k_{0}t$	0.0702	-	0.7284	0.0032	-	0.3945
First- order	$\frac{M_{t}}{M_{\infty }}=1-e^{-k_{1}t}$	0.2253	-	0.9191	0.0096	-	0.9744
Higuchi	$\frac{M_{t}}{M_{\infty }}=k_{H}\sqrt{t}$	0.2882	-	0.9401	0.0557	-	0.9689
Korsmeyer–Peppas	$\frac{M_{t}}{M_{\infty }}=kt^{n}$	0.315	0.206	0.9907	0.049	0.54	0.9995

Note: where *M_t_* is the cumulative mass of an active agent in the receiving medium over time *t*; *M_∞_* is the total mass of the active agent in the receiving medium at time *∞*; *k*_0_*, k*_1_, *k*_H_ and *k* are the release constants for the zero-order, first-order, Higuchi, and Korsmeyer–Peppas models, respectively; and *n* is the release exponent.

**Table 4 polymers-14-03493-t004:** Results of the survival assay and agar diffusion assay, which are reported as the average ± standard deviation.

	% Reduction in *S. aureus* Population	Inhibition Area (cm^2^)
PLLA-vancomycin without drug-free topcoat	91.77 ± 0.42	19.71 ± 2.25
PLLA-vancomycin with drug-free topcoat	81.77 ± 9.57	12.97 ± 1.3
Positive control	100.00 ± 0.00	24.56 ± 4.96
Negative control	Increase in the number of CFU	No inhibition zone

## Data Availability

Data presented in this study are available on request from the corresponding author.

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
