# Peer review of "Controlled Release Mechanism of Vancomycin from Double-Layer Poly-L-Lactic Acid-Coated Implants for Prevention of Bacterial Infection"

_polymers, 2022, doi:10.3390/polym14173493_

Round 1

Reviewer 1 Report

In this paper, an antimicrobial PLA coating has been designed to achieve controlled release of vancomycin. Although present study is well designed and shows a direct application in implantology, several concerns can be found.

1- Details about sterilization process of the samples must be provided: number of washes in ethanol, volume, time.. etc

2- How exactly residual bacterial suspension was recovered in survival assay after incubation? ATCC number should be also provided in this paragraph.

3- Bacterial biofilm is a mature structure with a complex extracellular matrix. In this paper, only first stages of biofilm formation has been observed. For this reason, the term “biofilm” should be avoided, and other proper synonyms such as “bacterial culture” could be used.

4- Although antimicrobial assays have been successful, in order to describe the early efficacy of surfaces avoiding bacteria proliferation a Live/Dead assay over samples would be welcome. Moreover, as many drug release surfaces based on PLA can be found in bibliography, an additional experiment using another cell line representing Gram negative lineage would improve its possible range.

5. In biocompatibility assay using mammalian cells, authors have only checked cytotoxicity of soluble extracts. For this assay, micrographs of cell culture after incubation with extracts should be provided to evaluate cell culture state. Moreover, as authors claim the usefulness of this materials in implantology, proliferation over PLA surfaces should be evaluated. In addition. including another complementary technique such as DNA content analysis would be necessary for this purpose.

6. Figure 3 has not a proper quality. English grammar and minor typos should be checked.

Reviewer 2 Report

This manuscript has given a controlled release method of Vancomycin from double-layer PLLA coated Implants. However, the novelty and content are not a bit enough, and some work should be improved and some problems should be clarified before publishing.

1.     Figure.3 should be given with high resolution. The font sizes in Figures.1,2,4 were too small.

2.     There is not Double-Layer structure observed in the SEM pictures.

3.     It will be more understandable if there is an illustration on preparation method and afforded double layered film.

4.     What is mechanism of forming honeycomb structure, which should be given more explanation. And what do preparation conditions affect the honeycomb structure, for example, the pore size, contact angle, specific surface area, etc.

5.     It is suggested that the Discussion should be followed with the resulting description, and not a separate section.

6.     Why do the authors use dip-coating and how to control the film thickness? If the steel was simulated as bone, how about the release direction of Vancomycin?

7.     How to solve the hydrophobicity problem of this film?

8.     If PMMA is being used in the current medical treatment, after similar dip-coating, the comparison should be given with PLLA.

9.     The controlled release time was extended 3 weeks, and is it a good release time for medical treatment?

10.  Except for positive bacteria, S. aureus, antibacterial activity of this film may be considered for another common negative bacterium, E. coli.

11.  The sentence should be improved in line 391-392 and there’re too many “with” or “without”.

12.  The main novelties and conclusions should be more concise and clear in the parts of Abstract and Conlusion.

Round 2

Reviewer 1 Report

Main concerns have been solved and manuscript can be published,

Reviewer 2 Report

Agree to publishing before carefully proofing.